# OpenReview forum: "PRISM: Robust VLM Alignment with Principled Reasoning for Integrated Safety in Multimodality"
_ICLR.cc/2026/Conference — Submitted to ICLR 2026_

### Official Review · Reviewer_Z9M5 · 2025-10-29

**Soundness:** 2
**Presentation:** 3
**Contribution:** 2
**Rating:** 4
**Confidence:** 4

**Summary:**

This paper proposes PRISM, a safety alignment framework for vision-language models (VLMs). The goal is to make models (1) refuse unsafe multimodal requests and (2) still answer benign queries helpfully instead of over-refusing. The key idea is to force the model to reason explicitly about safety in a structured 4-step format: (i) analyze the user's textual intent, (ii) describe the image in context, (iii) jointly reason about text and image to detect “problem-unsafe”, “image-unsafe”, or “combination-unsafe” risks, and (iv) either refuse with an explanation of which policy is violated or provide a helpful answer if it is safe. The authors also build PRISM-CoT (a supervised dataset of such structured reasoning traces) and PRISM-DPO (a preference dataset created by sampling multiple candidate reasoning steps and labeling safer/helpful steps as preferred). The final model is trained with supervised fine-tuning on PRISM-CoT and then Direct Preference Optimization using PRISM-DPO. Experiments on jailbreak benchmarks and adaptive attack settings show lower attack success rate while mostly keeping helpfulness.

**Strengths:**

1. The paper targets an important and realistic safety problem for multimodal models: jailbreaks where the harmful intent is only visible when combining the image and the text. This “combination-unsafe” setting is largely underexplored in prior work and is clearly defined here.
2. The use of a fixed 4-step reasoning template (Problem, Caption, Reasoning, Output) is conceptually interesting. It encourages the model to explicitly articulate why something is unsafe, instead of giving a generic refusal, and it tries to avoid over-defensive behavior on benign inputs.
3. The training pipeline explicitly balances safety and helpfulness. The data includes both malicious and benign cases, and “good” behavior is either (i) a clear, policy-grounded refusal for unsafe queries or (ii) an actually helpful answer for benign queries, not blanket denial.
4. The evaluation is broad. The paper reports results on multiple multimodal jailbreak benchmarks, includes an adaptive attacker setting, and also measures helpfulness. Reported numbers suggest PRISM reduces attack success without collapsing utility.

**Weaknesses:**

1. The “MCTS” component in PRISM-DPO appears oversold. The paper claims to use Monte Carlo Tree Search to build preference data. But what is actually done is: for each reasoning step, the model samples a few candidate continuations (e.g., k=3k=3k=3), GPT-4o scores each candidate for safety (on malicious inputs) or helpfulness (on benign inputs), and the higher-scoring candidate is treated as the positive example while the lower-scoring one is treated as negative. Safety rewards are not back-propagated to earlier steps, and preference pairs are formed locally at the same depth using simple score thresholds. This is essentially per-step sampling plus reranking. It is not clear that UCB, visit counts, or other MCTS machinery is doing meaningful global search over full 4-step trajectories. As written, “safety-aware MCTS” looks more like branding than a necessary algorithmic ingredient.
2. There is no ablation to justify that this “MCTS”-based data construction is actually better than a simpler baseline. The paper does not compare (i) PRISM-CoT SFT only, (ii) SFT with naive DPO built from per-step top-vs-bottom scoring without any tree search framing, and (iii) full PRISM with the claimed MCTS loop. Without this, it is hard to judge how much of the reported gains actually come from the proposed DPO construction versus just standard preference tuning.
3. The whole pipeline depends on GPT-4o as the judge for both safety (unsafe queries) and helpfulness (benign queries), and the final evaluation also relies on automated judges. This raises questions about potential bias inherented from the model, the concerns on reproducibility, and whether the safety the model learns is simply “whatever GPT-4o calls acceptable” rather than a stably safe policy.

**Questions:**

N/A

---

> ### Author Response · Authors · 2025-11-25
> **Rebuttal to Reviewer Z9M5 (Part 1/2)**
>
> We thank the reviewer for recognizing that our "combination-unsafe" setting is "largely underexplored" and that our fixed 4-step reasoning template is "conceptually interesting" for avoiding over-defensive behavior. We address the reviewer's specific concerns in detail below.
>
>
> **Ablations among (i) SFT only, (ii) naive per-step DPO, (iii) full PRISM.** We agree with the reviewer that a rigorous three-way comparison is essential to justify the benefits of our structured pipeline over simpler baselines.
>
> To make the setting precise, we evaluate three training configurations:
> - **SFT only:** supervised fine-tuning on four-stage formatted data to teach the model to follow the structured reasoning template, no preference optimization.
> - **SFT + naive per-step DPO:** at each stage, the model generates four steps without any tree structure or branching, and we apply the same stage-local reward rules used by PRISM directly to these steps.
> - **SFT + PRISM-DPO:** structured stage-wise tree search with per-stage branching.
>
> | Training Method           | JailbreakV-mini (ASR) ↓ | MIS-hard (ASR) ↓ | MM-Vet-v2 (Acc) ↑ |
> | ------------------------- | -------------- | --------------- | --------------- |
> | SFT only                  | 15.0    |  39.5           | 37.3            |
> | SFT + naive per-step DPO  | 8.8    |  9.6           | 33.2            |
> | SFT + PRISM-DPO           | 0.41    |   6.3           | 41.3            |
>
> Interpreting the results, SFT only provides modest safety improvements while preserving utility because it primarily enforces adherence to the four-stage format. Naive per-step DPO attains lower ASR yet reduces MM-Vet-v2 accuracy, consistent with over-defense. Without careful rewarding, the learner struggles to capture nuanced differences when penalizing unsafe fragments, often leading to over-defensive behavior (i.e., refusing benign questions). PRISM-DPO delivers both the lowest ASR and the highest utility, confirming the benefit of structured, stage-local search over standard per-step preference optimization. This empirical evidence justifies the structured data construction over standard preference optimization.

---

> > ### Author Response · Authors · 2025-11-25
> > **Rebuttal to Reviewer Z9M5 (Part 2/2)**
> >
> > **"MCTS" framing vs. Per-step sampling.** We thank the reviewer for their observation regarding the "MCTS" terminology.
> >
> > We intentionally deviate from traditional MCTS designs due to the unique characteristics of our method. Because PRISM employs a structured four-stage reasoning process, we have access to ground-truth supervision at each individual step. This allows us to compute **per-step rewards**, a crucial difference from classical MCTS settings where only the terminal reward is known and must be backpropagated through all preceding steps.
> >
> > We **intentionally do not backpropagate safety rewards** through the reasoning tree. As discussed in the paper, malicious intent may appear at any stage, and propagating safety penalties to earlier, non-malicious reasoning steps would inappropriately attribute blame and distort training, leading to the problem of **over-defense** and utility degradation on benign tasks.
> >
> > In addition, **our helpfulness rewards are fully backpropagated**, following classical MCTS credit assignment and enabling global optimization of reasoning quality on benign inputs. This backpropagation of helpfulness scores, combined with their use in branching, increases sampling efficiency and improves utility compared to a purely per-step optimization baseline.
> >
> > By combining these careful design choices for safety and helpfulness, PRISM can more accurately penalize safety violations with minimal harm on utility. The ablation results in the table above demonstrate that our approach consistently outperforms the baseline, confirming that the structured search process and differential reward strategy indeed add significant value. We will refine our paper to ensure this distinction and rationale are explicitly clear.
> >
> > **Dependence on GPT-4o judge; bias and reproducibility concerns.** We appreciate the reviewer's valid concern regarding judge dependence and fully agree that ensuring reproducibility regardless of the model bias. **Judge Robustness Experiment:** Our reward scoring is rule-based and anchored to ground-truth labels (e.g., explicit safety policy violations for unsafe queries, or gold reasoning steps for benign data), rather than opaque model preferences. To further demonstrate the absence of model-specific bias, we measured the pair-wise alignment rate of preference scoring using four diverse models: GPT-4o, GPT-4o-mini, Qwen3-VL-8B, and Gemini-2.5-Flash. The high agreement across all model pairs confirms that the safety signals are robust and evaluator-agnostic.
> >
> > | Model       | GPT-4o | GPT-4o-mini | Qwen3-VL | Gemini-2.5 |
> > | ----------- | ------ | ----------- | -------- | ---------- |
> > | GPT-4o      | 100.0% | 97.1%       | 92.4%    | 94.0%      |
> > | GPT-4o-mini | 97.1%  | 100.0%      | 92.0%    | 88.5%      |
> > | Qwen3-VL    | 92.4%  | 92.0%       | 100.0%   | 90.2%      |
> > | Gemini-2.5  | 94.0%  | 88.5%       | 90.2%    | 100.0%     |

---

> ### Comment · Reviewer_Z9M5 · 2025-11-27
>
> Thank you for the detailed response. I have carefully reviewed the additional results and explanations, and they satisfactorily address my earlier concerns. These clarifications improve my confidence in the soundness of the approach and the validity of the conclusions. Accordingly, I have decided to raise my score.

---

> > ### Author Response · Authors · 2025-11-27
> >
> > Thank you very much for reviewing our rebuttal and for raising your score. We are encouraged to hear that our response resolved your concerns. We will ensure that the new clarifications and further ablations are included in the final version to further enhance its quality. We truly appreciate the time and effort you dedicated to improving our paper.

---

### Official Review · Reviewer_425i · 2025-10-29

**Soundness:** 3
**Presentation:** 3
**Contribution:** 2
**Rating:** 4
**Confidence:** 4

**Summary:**

This paper proposes PRISM framework to safeguard vision-language models. The framework contains two stages: (1) PRISM-COT, using GPT-4o to generate step-by-step safety-related reasoning traces and reformat parts of benign data from LLAVA-CoT to construct the SFT dataset. (2) leveraging MCTS with the designed safety reward and helpful reward to construct preference data, then perform DPO training. Extensive experiments demonstrate the effectiveness of PRISM in enhancing the safety performance of VLMs and preserving model utility.

**Strengths:**

1. The paper is well organized and clear
2. The performance of PRISM is strong, both in safety evaluation and utility

**Weaknesses:**

1. **Novelty:** The proposed PRISM is too similar to STAIR[1], where the only difference is modality.
   - Both PRISM and STAIR are two-stage training frameworks, including the long step-by-step reasoning SFT, and an MCTS-based DPO.
   - The reward design in the MCTS stage is limited compared to STAIR. STAIR combines safety and helpful rewards together instead of safety reward for safety-related queries, and helpful reward for general verifiable queries.
   - STAIR can perform self-improvement during the DPO stage without the external signal, which is hard to perform in PRISM-DPO.
   - Could the authors clarify the differences between PRISM and STAIR beyond the modality aspect?

2. **Can PRISM be extended to online-RL?**
   - During the DPO stage, the preference data is annotated based on safety reward and helpful reward. Is it able to perform online-RL (PPO, GRPO) to enhance model performance? Since it is a more common method to enhance model reasoning capability.

3. **More experiments**
   - Can the author provide results on more general VQA benchmarks to prove PRISM preserves the utility (e.g. MMMU, MMStar)
   - The test-time scaling is applied only to the safety task. However, it is a common strategy that can also be employed for general tasks. It would be better to evaluate test-time scaling on benchmarks such as MM-Vet, MMMU, and MMStar as well.

[1] STAIR: Improving Safety Alignment with Introspective Reasoning

**Questions:**

See Weakness

---

> ### Author Response · Authors · 2025-11-25
> **Rebuttal to Reviewer 425i (Part 1/2)**
>
> We thank the reviewer for finding the paper "well organized" and acknowledging that the performance of PRISM is "strong, both in safety evaluation and utility." We have provided detailed responses to the reviewer's questions below.
>
> **Q1: Novelty relative to STAIR.** We thank the reviewer for this insightful comparison. While PRISM shares the high-level two-stage (SFT+DPO) paradigm with STAIR, we wish to clarify that extending this framework to the multimodal domain requires addressing fundamentally distinct challenges. PRISM introduces three key architectural innovations specifically designed to resolve these multimodal complexities, which go beyond a simple change of input modality.
>
> - **Structural Adaptation to Multimodal Threats** While STAIR effectively handles text-only threats, the vision-language domain introduces a unique threat model: Problem Unsafe, Image Unsafe and their combination unsafe where benign text and image components combine to form a harmful intent. To address this, PRISM enforces a Structured 4-Stage Reasoning (Problem $\to$ Caption $\to$ Reasoning $\to$ Output) derived explicitly from a multimodal safety taxonomy. This structure is essential for disentangling visual context from textual intent, a capability that requires explicit architectural enforcement rather than the free-form reasoning used in text-only models. And we have conducted a controlled ablation by perturbing each of the four stages via appending end tokens to skip specific steps. Results in the table below prove that defining the structured reasoning process is key to success, which highlights our contribution through a carefully designed structured reasoning procedure.
>
> | Setting               | MIS-hard (ASR) ↓ | MMMU-Pro (Acc) ↑ |
> | --------------------- | ---------------- | ---------------- |
> | Full PRISM-DPO        | 11.3             | 41.27            |
> | w/o PROBLEM           | 19.3             | 39.42            |
> | w/o CAPTION           | 14.5             | 40.00            |
> | w/o PROBLEM + CAPTION | 29.5             | 36.07            |
> | w/o REASONING         | 20.0             | 38.21            |
>
>
> - **Efficacy of Stage-Local Rewards** Our analysis suggests that for VLMs, Stage-Local Blame Assignment is critical for balancing safety and utility. In a multimodal context, a model might caption an image correctly (Step 2) but fail in safety reasoning (Step 3). A monolithic reward would penalize the entire trajectory, potentially degrading the visual perception performance. PRISM's stage-local approach ensures we penalize only the specific error, thereby preserving the model's visual understanding capabilities.
> - **External Oversight vs. Self-Improvement** Regarding self-improvement, while internal signals function well for objective reasoning tasks, safety alignment often benefits from System 2 oversight. Our proposed PRISM utilizes MCTS with an external objective rooted in our safety taxonomy. This provides a rigorous verification signal that is particularly valuable when defining complex multimodal safety boundaries.
>
> Additionally, we also conduct a head-to-head comparison with STAIR. For a head-to-head comparison, we faithfully re-implement the vanilla STAIR pipeline and train STAIR-1k on the identical 1k subset used for PRISM-1k. While both methods achieve comparable accuracy on the benign MMMU-Pro benchmark, the safety gap is striking: STAIR-1k lags far behind on MIS-hard, where neither text nor image alone reveals safety violation. This contrast underscores that PRISM’s structured, stage-wise reasoning is important for uncovering subtle multimodal safety violations, which is precisely the challenge our method is designed to master.
>
> | Models             | MMMU-Pro Acc| MIS-hard ASR| VLBreak ASR|
> | ------------------ | ------------| ----------- | ---------- |
> | Qwen2-VL (Base)    |    44.2     |    78.02    |    15.12   |
> | + STAIR-1k         |    41.4     |    33.19    |    3.19    |
> | **+ PRISM-1k**     |    42.9     |    13.00    |    0.88    |

---

> > ### Author Response · Authors · 2025-11-25
> > **Rebuttal to Reviewer 425i (Part 2/2)**
> >
> > **Q2: Extension to online RL (PPO/GRPO).** We sincerely appreciate this thoughtful suggestion and agree that online RL is a promising avenue to further enhance reasoning. In the present work, our core focus is on achieving multimodal safety with minimal utility impact via structured reasoning, making algorithms like PPO or GRPO beyond our current scope. We instead leveraged DPO, which effectively optimizes preferences without requiring explicit reward model training [1]. Looking ahead, we believe future work should explicitly co-optimize safety and utility to foster responsible VLM development. We will add a dedicated discussion note outlining how stage-local rewards and utility constraints can be integrated into PPO/GRPO, which we view as an exciting direction.
> >
> > [1] Rafailov, R., Sharma, A., Mitchell, E., Ermon, S., Manning, C.D., & Finn, C. (2023). Direct Preference Optimization: Your Language Model is Secretly a Reward Model. ArXiv, abs/2305.18290.
> >
> > **Q3: General VQA and Test-Time Scaling.** We thank the reviewer for the excellent suggestion to validate our utility preservation on broader, general benchmarks. As suggested, we evaluated PRISM on MM-Vet-v2, MMMU-Pro. PRISM maintains competitive utility against the strong Qwen2-VL baseline while significantly outperforming other safety-aligned models (e.g., VLGuard).
> >
> > | Models             | MM-Vet-v2 | MMMU-Pro |
> > | ------------------ | --------- | -------- |
> > | Qwen2-VL (Base)    | 54.4      | 44.2     |
> > | + SafeRLHF-V       | 12.9      | 38.7     |
> > | + SPA-VL           | 46.8      | 40.6     |
> > | + VLGuard          | 17.7      | 26.5     |
> > | **+ PRISM (Ours)** | 48.9      | 41.2     |
> >
> >
> > We also agree that Test-Time Scaling is applicable to general tasks and have added relevant evaluations. The table below reports test-time scaling on a subset of MM-Vet-v2 and MMMU-Pro under compute budgets 1, 2, 4, 8, and 16.
> >
> > | Compute Budget | MM-Vet-v2 (Acc) ↑ | MMMU-Pro (Acc) ↑ |
> > | -------------- | ----------------- | ---------------- |
> > | 1              | 45.6                 | 42.0                |
> > | 2              | 50.0                 | 43.1                |
> > | 4              | 49.5                 | 44.0                |
> > | 8              | 52.1                 | 44.0                |
> > | 16             | 52.5                 | 46.2                |
> >
> > We respectfully note that benign utility improvements from test-time scaling are generally more modest than the corresponding safety gains. This outcome aligns with PRISM’s primary objective: to strengthen multimodal safety while maintaining minimal impact on utility.

---

> > > ### Author Response · Authors · 2025-11-27
> > > **Kind Follow-up on Our Rebuttal**
> > >
> > > Dear Reviewer 425i,
> > >
> > > We hope you are doing well. We wanted to kindly let you know that we have submitted our rebuttal addressing all of your questions in detail. We would be truly grateful if you could take a look and let us know whether there are any remaining issues we should clarify.
> > > We very much appreciate the time and effort you have already invested, and we sincerely thank you for considering our responses when forming your final evaluation.
> > >
> > >
> > > Warm regards,
> > >
> > > The Authors

---

### Official Review · Reviewer_R53w · 2025-11-07

**Soundness:** 2
**Presentation:** 2
**Contribution:** 2
**Rating:** 4
**Confidence:** 2

**Summary:**

The paper introduces PRISM, a reasoning-based safety alignment framework for Vision–Language Models. It constructs (i) PRISM-CoT, a four-stage safety-aware reasoning dataset (Problem, Caption, Reasoning, Output), and (ii) PRISM-DPO, a step-level MCTS-generated preference dataset for Direct Preference Optimization.

**Strengths:**

- This paper addresses a timely and important challenge in multimodal safety by proposing a structured reasoning pipeline that explicitly decomposes and interprets image–text interactions responsible for jailbreaking.

- This paper builds a comprehensive multimodal safety dataset combining structured reasoning traces (PRISM-CoT) and MCTS-generated step-level preferences (PRISM-DPO), offering valuable supervision for fine-grained safety alignment.

**Weaknesses:**

- This paper does not include ablations or diagnostic experiments that isolate whether safety gains come from the structured reasoning process itself. Without controlled ablations that remove or vary reasoning steps, it is unclear whether improvements reflect genuine reasoning ability or additional supervision.

- Since GPT-4o is used for data generation, reward definition, and evaluation, the framework effectively learns to reproduce its evaluator’s preferences, forming a closed alignment loop that risks self-confirmation rather than genuine robustness.

- The experiments do not include recent VLMs such as Qwen3-VL or other latest models, and lack analysis across different model scales, limiting evidence of how well PRISM generalizes.

**Questions:**

- How can the authors demonstrate that PRISM’s improvements stem from structured reasoning rather than from additional supervision or data curation, for example through controlled ablations or reasoning-step perturbation studies?

- Given that GPT-4o is used throughout supervision and evaluation, how can the authors ensure that PRISM is learning evaluator-agnostic safety reasoning rather than merely aligning to GPT-4o’s own safety priors?

- Have the authors explored applying PRISM to more recent or larger-scale VLMs (e.g., Qwen3-VL or similar) to verify whether its structured reasoning framework scales consistently with model capacity?

---

> ### Author Response · Authors · 2025-11-25
> **Rebuttal to Reviewer R53w (Part 1/2)**
>
> We appreciate the reviewer's positive assessment that PRISM constructs a "comprehensive multimodal safety dataset" and offers "valuable supervision for fine-grained safety alignment." We address the specific questions raised by the reviewer below.
>
> **Q1: Ablations on isolating structured reasoning gains.** We thank the reviewer for highlighting the need to isolate whether gains stem from the structured reasoning process itself rather than just data curation. To address this, we performed controlled ablations by perturbing each of the four stages (Problem, Caption, Reasoning, Output) via appending end tokens to skip specific steps.
>
> | Setting               | MIS-hard (ASR) ↓ | MMMU-Pro (Acc) ↑ |
> | --------------------- | ---------------- | ---------------- |
> | Full PRISM-DPO        | 11.3             | 41.27            |
> | w/o PROBLEM           | 19.3             | 39.42            |
> | w/o CAPTION           | 14.5             | 40.00            |
> | w/o PROBLEM + CAPTION | 29.5             | 36.07            |
> | w/o REASONING         | 20.0             | 38.21            |
>
>
> As shown in the table above, the full PRISM pipeline achieves the best balance. Notably, the *Reasoning* stage is crucial; removing it causes a significant drop in benign utility (38.21 vs 41.27), confirming that the improvements reflect genuine reasoning ability rather than just data curation. On MIS-hard, full PRISM attains the lowest ASR (11.3). Ablating any stage increases ASR, which is most notable when both Problem and Caption are removed (29.5). This pattern indicates that PRISM's structured stages provide complementary signals for localizing and suppressing unsafe trajectories. This experiment further verifies that defining the structured reasoning process is key to success, highlighting the contribution of our paper through a carefully designed structured reasoning procedure.

---

> > ### Author Response · Authors · 2025-11-25
> > **Rebuttal to Reviewer R53w (Part 2/2)**
> >
> > **Q2: Concerns about closed alignment loop (GPT-4o judge).** We appreciate the reviewer raising this concern regarding potential closed-loop alignment. We would like to clarify that GPT-4o is used for **Cold Start & Guided Scoring**. SFT annotations primarily serve as a cold start to reformat raw data into our four-stage structure. The subsequent reward scoring is rule-based and anchored to ground-truth labels (e.g., explicit safety policy violations for unsafe queries, or gold reasoning steps for benign data), rather than opaque model preferences.
> >
> > To further prove its robustness, we substitute Qwen2.5-VL for GPT-4o as the SFT data curator, and this yields highly comparable post-DPO performance (first table below).
> >
> > | SFT Data Curator          | MIS-hard (ASR) ↓ | MMMU-Pro (Acc) ↑ |
> > | ------------------------- | -------------- | --------------- |
> > | GPT-4o                    | 6.34           | 41.3            |
> > | Qwen2.5-VL (Self-Curated) | 7.59           | 40.9            |
> >
> >
> > To further demonstrate evaluator-agnostic scoring, we measured pair-wise alignment rates of preference signals across four diverse models (second table). The high agreement across all model pairs confirms that the safety signals are robust and not dependent on a specific judge model.
> >
> > | Model       | GPT-4o | GPT-4o-mini | Qwen3-VL | Gemini-2.5 |
> > | ----------- | ------ | ----------- | -------- | ---------- |
> > | GPT-4o      | 100.0% | 97.1%       | 92.4%    | 94.0%      |
> > | GPT-4o-mini | 97.1%  | 100.0%      | 92.0%    | 88.5%      |
> > | Qwen3-VL    | 92.4%  | 92.0%       | 100.0%   | 90.2%      |
> > | Gemini-2.5  | 94.0%  | 88.5%       | 90.2%    | 100.0%     |
> >
> > **Q3: Missing recent VLMs and scale analysis.** We appreciate the reviewer's suggestion to evaluate on the latest models to demonstrate scalability. We kindly note that **Qwen3-VL** was **not released** at the time of our submission. However, we acknowledge the importance of staying current with the state-of-the-art, and we will do our best to incorporate results from Qwen3-VL with different scales into the final version of our paper.

---

> ### Author Response · Authors · 2025-11-27
> **Kind Follow-up on Our Rebuttal**
>
> Dear Reviewer R53w,
>
> We hope you are doing well. We wanted to kindly let you know that we have submitted our rebuttal addressing all of your questions in detail. We would be truly grateful if you could take a look and let us know whether there are any remaining issues we should clarify. We very much appreciate the time and effort you have already invested, and we sincerely thank you for considering our responses when forming your final evaluation.
>
> Warm regards,
>
> The Authors

---

### Meta-Review · Area_Chair_psnr · 2026-01-12

**Summary:**

This paper proposes PRISM framework to safeguard vision-language models.

Reviewers main concerns are about novelty, extension of PRIMS to online-RL, and generalizability of the experiments.

- Novelty: The proposed PRISM is too similar to STAIR[1], where the only difference is modality.
- The experiments did not adequately show if the proposed method preserves model utility.
- The whole pipeline depends on GPT-4o as the judge for both safety (unsafe queries) and helpfulness (benign queries), and the final evaluation also relies on automated judges.

**Reviewer Concerns:**

Some of the reviewer concerns were partially addressed during the rebuttal. For instance, to address the dependence on GPT-4o as a judge, authors provided additional experiments using GPT-4o-mini, Qwen3-VL-8B, and Gemini-2.5-Flash. Regarding Novelty, authors argued that their method extends safety-aware chain of thought to multimodal domain that requires structural adaptation.

Overall, the nature of comments and additional experiments suggest that the paper should go through a major revision and another round of careful review to verify all the claims.

**Reviewer Scores:**

Reviewers scores were 4,4,4

While one reviewer increased the score in a comment, it is unclear if other reviewers would change their scores.
I feel the nature of concerns warrant a major revision and a careful review.

---

### Decision · Program_Chairs · 2026-01-26

Reject